# Towards Mitigating Jellyfish Attacks Based on Honesty Metrics in V2X Autonomous Networks

Messaoud Benguenane [1,*], Ahmed Korichi [1], Bouziane Brik [2] and Nadjet Azzaoui [1]

[1] LINATI Laboratory, Department of Computer Science and Information Technology, Faculty of New Technologies of Information and Communication, University of Kasdi Merbah, Ouargla 30000, Algeria
[2] Laboratory DRIVE EA 1859, University Bourgogne Franche Comté, 58000 Nevers, France
* Correspondence: benguenane@gmail.com

**Abstract:** In vehicle-to-everything (V2X) networks, security and safety are inherently difficult tasks due to the distinct characteristics of such networks, such as their highly dynamic topology and frequent connectivity disruptions. Jellyfish attacks are a sort of denial of service attack that are challenging to deal with, since they conform to protocol norms while impairing network performance, particularly in terms of communication overhead and reliability. Numerous existing approaches have developed new techniques with which to identify and prevent these attacks; however, no approach has been capable of facing all three types of Jellyfish attacks, which include reordering attacks, delay variance attacks, and periodic drop attacks. In this work, we design a new protocol that analyzes the behavior of every node in a network and selects the trusted routes for data transmission to their intended destination by calculating different Honesty metrics. The OMNET++ simulator was used to evaluate the overall performance of the proposed protocol. Various evaluation metrics, such as the packet delivery ratio, end-to-end delay, and throughput, are considered and compared with other existing approaches.

**Keywords:** V2X networks; security and safety; Jellyfish attacks; secure routing protocol





## 1. Introduction

V2X (vehicle-to-everything) networks are wireless communication networks that enable vehicles to communicate with other vehicles, infrastructures, pedestrians, and devices in their vicinity. These networks are a critical component of intelligent transportation systems (ITSs) and are expected to play a significant role in the development of autonomous vehicles [1–3]. V2X networks can use different types of wireless technologies, such as dedicated short-range communications (DSRCs), cellular vehicle-to-everything (C-V2X). DSRCs use a dedicated frequency band, while C-V2X uses existing cellular networks to enable communication between vehicles and the surrounding infrastructures [4,5]. V2X networks can provide a wide range of services, such as collision avoidance, traffic management, and cooperative driving. For example, V2X technology can alert drivers to potential collisions, provide real-time traffic information, and enable vehicles to work together to optimize traffic flow. In addition to improving safety and efficiency on the road, V2X networks can also reduce traffic congestion, reduce emissions, and enhance the overall driving experience. As a result, V2X networks are expected to become increasingly important in the future of transportation [6,7].

Routing in V2X networks is the process of selecting the best path for transmitting data between vehicles, infrastructures, and other devices in a network. The goal of routing in V2X networks is to ensure that data are transmitted efficiently, reliably, and securely [8]. There are several routing protocols that can be used in V2X networks. Topology- and location-based routing protocols are the two main categories of routing protocols in V2X networks [9]. Due to the high mobility of vehicles, location-based (or geographic) protocols

also outperform traditional ad hoc routing protocols [10,11]. The most important aspect of vehicle knowledge is location. Every node is effectively required to keep track of both its own location and that of its neighboring nodes [12]. In order to find a route to a destination, geographic routing systems use the geographic coordinates offered by a positioning system, such as the geographic positioning system (GPS) [13]. Among these protocols is the greedy perimeter stateless routing protocol (GPSR) [14], which is one of the most well-known and researched geographic routing methods. It assumes that every node is aware of its own location coordinates and periodically broadcasts them to its one-hop neighbors. GPSR uses perimeter forwarding and greedy forwarding strategies to route packets to their intended destinations. The next-hop neighbor node closest to the destination is chosen by the forwarding mechanism using a greedy algorithm. When greedy forwarding fails to route packets, perimeter forwarding will be used to forward packets [15].

In this context, security is a critical aspect of routing in V2X networks. Since V2X networks transmit sensitive information, such as location data and vehicle status, it is important to protect the confidentiality, integrity, and availability of this information. Traditional solutions, such as the use of encryption and decryption, may be able to prevent external malicious attackers from accessing sensitive data on networks; however, networks can be attacked by legitimate members, making nodes unable to protect communication from internal threats [16,17]. Networks can be easily disrupted by attackers through jamming, causing congestion, manipulating network routes, delaying or reordering packets, etc. In fact, denial of service (DoS), Wormhole attacks, Blackhole attacks, Grayhole attacks, and Rushing attacks are just some of the attacks that might disrupt networks [18,19]. Jellyfish attacks are a specific class of DoS attacks that are hard to detect and have a negative impact on overall network performance [20]. Specifically, they consist of introducing more delays in the transmission control protocol (TCP) and UDP packet transmission, where attackers can scramble packet orders before forwarding them to the destination node.

To deal with Jellyfish attacks and other attacks of a similar nature, a variety of security mechanisms have been proposed; however, these security mechanisms are not effective against all three types of Jellyfish attacks, such as Jellyfish reorder attacks, Jellyfish periodic drop attacks, and Jellyfish delay variance attacks, because they do not consider their impact. To guarantee the successful prevention of various Jellyfish attack categories, we need to choose the appropriate metric for every Jellyfish category based on their impact.

In this paper, we design a new security protocol to deal with Jellyfish attacks on top of the greedy perimeter stateless protocol (GPSR) as a routing protocol [14]. Thus, we propose a new secure and efficient routing protocol for preventing Jellyfish attacks in V2X, called SecE-V2X. In order to defend networks against different Jellyfish attack categories, our proposed protocol selects the trusted paths for routing packets to their destinations by calculating the Honesty criterion. This latter factor is determined based on three metrics: the packet delay, packet loss rate, and packet reordering. A node's Honesty value is dependent upon each node's properties and endorsements from its neighbors.

The remainder of this paper is organized as follows: Section 0 describes Jellyfish attacks and their variants. Related works and attack protection mechanisms are discussed in Section 0. Section 0 describes the proposed SecE-V2X routing protocol. The simulation results are provided in Section 0. Section 0 provides conclusions and some perspectives. Finally, Section 0 describes the adopted abbreviations.

## 2. Jellyfish Attack Overview

According to the authors of [21], Jellyfish attacks are described as three types of DoS attacks against mobile ad hoc networks (MANETs). They are classified as passive attacks, since they are launched from a network itself and ensure that both control and data protocols are compliant to make detection and protection extremely difficult tasks. By dropping, delaying, or reordering packets, the main goal of Jellyfish attacks is to reduce all flows' throughputs and create congestion, which degrade network performance. Jellyfish attacks are categorized into three subcategories: Jellyfish reorder attacks, Jellyfish periodic

drop attacks, and Jellyfish delay variance attacks. Table 1 shows a comparison between different jellyfish attack categories.

**Table 1.** Comparison of jellyfish attack categories.

| Jellyfish Attack Subcategory | Aim | Impact |
|:---:|:---:|:---:|
| Reordering attack | Scrambling the packet order | Decreases throughput<br>Increases network congestion |
| Periodic drop attack | Discarding packets periodically | Data are lost in communication<br>Network throughput falls |
| Delay variance attack | Introduces delay before transmitting | Increase delay<br>Significant network congestion |

### 2.1. Jellyfish Reordering Attacks

As the term implies, the attacker node ignores the first in, first out (FIFO) queue's basic function of data forwarding and instead selects packets at random before transmitting them to its next hop node [21], as demonstrated in Figure 1. Because some of the reordered packets' acknowledgments (ACKs) were not received in a timely manner, the sender had to retransmit them. When a packet is received, the receiver automatically generates an ACK for the packet. In the case of out-of-order packets, the sender will receive redundant ACK messages. If the number of duplicate ACK messages reaches a certain threshold level, the TCP will launch its flow control mechanism, which decreases the overall throughput and causes the network to become more congested [22].

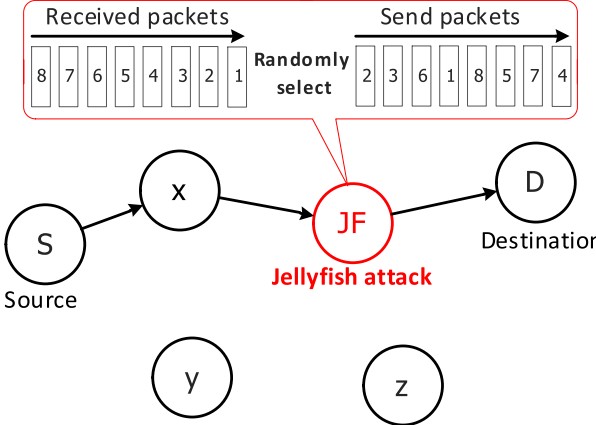

**Figure 1.** Jellyfish reordering attack.

### 2.2. Jellyfish Periodic Drop Attacks

In this attack, the malicious node deliberately selects and discards a portion of the packets or all of the packets for a short period of time in the midst of the ongoing communication process, as illustrated in Figure 2. Moreover, the ACK packets should arrive at the source node before the retransmission timeout (RTO) value expires, but they are delayed due to a periodic dropping attack, leading the source to believe that the packet has been destroyed [23]. This timeout is interpreted by the sender as a sign of extreme congestion, and the slow start phase is initiated. The RTO value is compounded for each retransmission of unacknowledged segments until it reaches a threshold RTO value, at which point the connection is terminated [24]. As the frequency of packets discarded by the attacker node increases the network throughput decreases. A Jellyfish node may delete packets as soon as the sender exits its slow start phase to maximize the damage of attacks. As a result, the flow will constantly be in a precarious slow start state [25].

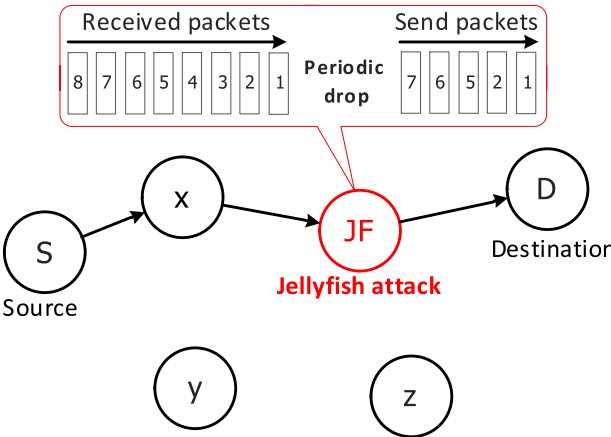

**Figure 2.** Jellyfish periodic drop attack.

### 2.3. Jellyfish Delay Variance Attacks

In a delay variance attack, the malicious nodes are selfishly delaying packets. As shown in Figure 3, the impacted Jellyfish node maintains the FIFO queue's rules while forwarding data packets, but it introduces some random delay before transmitting them to their destination without scrambling the packet order. In this scenario, the sender will not receive an ACK within the time frame given, which makes the source node assume that packets have been lost and will begin retransmitting them [26]. Because there is no other way to detect lost packets due to significant network congestion, managing network traffic becomes extremely challenging [24].

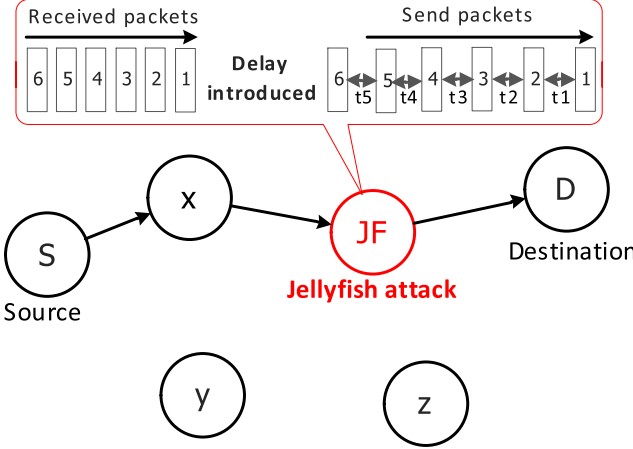

**Figure 3.** Jellyfish delay variance attack.

## 3. Related Literature

The primary purpose of a secure vehicular ad hoc network is to provide secure and successful data transmission between sources and destinations. It is necessary to develop a security mechanism that can make a network resistant to different attacks in order to perform the network efficiently.

Attacks such as Wormhole, Blackhole, and Sybil attacks seem to be very easily recognizable because the attacker node does not employ any protocol instructions and disrupts the routing protocol's operating conditions by inserting bogus data, modifying information, and removing data in control packets [27,28]. Attacks such as Jellyfish attacks, on the other hand, are more difficult to detect since they respect the variety of protocol standards and disrupt normal network operations [29].

Simulation research examining the impact of Jellyfish attacks clearly reveals that they have a significant influence on throughput and end-to-end delay [30]. To detect and mitigate the malicious activities of different Jellyfish attack variations, the research community has put significant effort into developing detection and countermeasure approaches, but so far no approach has been developed to protect against all three types of Jellyfish attacks together. A proposed solution is to identify Jellyfish attacks by using a timer to measure delay and then finding a different route as well as blocking the attacker node in order to avoid delay variance attacks and improve throughput [31]. A modified hash function and time–space cryptography are also used to determine whether packet reordering is being carried out as a result of a Jellyfish attack, and, if so, to prevent the attack and improve network performance [32].

Other strategies, based on trust and reputation, were considered effective security strategies for open environments such as MANETs [26,33]. To solve DoS/distributed denial of service (DDoS) attacks, a trust-based approach that depends on the history and profile of attacks is used to provide a solution for vehicular ad hoc network (VANET) security by Poongodi et al. [34]. The packet delivery ratio, delay, average latency, and energy usage are all metrics that are taken into account. According to their acquired data, it achieved a detection rate of 95.8%, an average delay of 30 s, and a packet delivery ratio of 86%. The authors in [35] are developing another trust-based collaborative intrusion detection system. This approach is also designed to protect availability services from DoS/DDoS attacks. In such trust-based situations, sophisticated attacks can still be launched.

Machine learning is a well-established approach for VANETs that has been proven to be particularly useful for attack prediction and analysis. Denial of service (DoS) and its variants are commonly prevented through the use of machine learning and deep learning techniques. The detection of DDoS attacks has been presented in [36] through the use of an efficient hybrid approach that is based on the support vector machine (SVM) kernel. Jitters, packet drops, collisions, and other features are employed in the simulation to generate various forms of data in modeling a real-time situation that contains characteristics of both regular and DDoS attacks. The suggested algorithm is used to evaluate the performance of the hybrid model in order to determine how well it can distinguish between regular communications and DDoS attacks.

The authors of [37] used an intrusion detection system (IDS) algorithm to prevent ad hoc on-demand distance vector (AODV) routing protocols against Jellyfish and Blackhole attacks in VANETs. They compare the conditions without attacks to those with Blackhole and Jellyfish attacks when analyzing the AODV routing protocol with and without an IDS algorithm. The results of the simulation show that the addition of the IDS algorithm increases the performance of the network based on the quality of service (QoS) parameters: packet delivery ratio (PDR), throughput, and end-to-end delay compared with conditions under attacks.

M. Kaur et al. [38] proposed utilizing a genetic algorithm to improve overall network performance in terms of latency, throughput, PDR, and energy efficiency for detecting and combating Jellyfish attacks in MANETs. This strategy is quite effective at providing a defense against Jellyfish's periodic dropping attacks.

In order to combat Jellyfish reorder attacks on MANETs, the authors of [39] presented a secure link formation mechanism. They investigated severe flaws and backdoors in multicast routing protocols and suggested a defense technique that is both secure and robust. The proposed approach improves the throughput and packet delivery ratio in MANETs, according to simulation results.

By designing a secure routing protocol, J. Soni and K. Uikey [40] proposed a security method to protect MANETs from Byzantine attacks. The reliability is improved and congestion is decreased by choosing secured multiple pathways with only the defective links (not the entire path) removed. The Byzantine faults are located using adaptive probe signals. When the loss rate goes above a certain point, searching for the attackers will begin. Next, the routes between the source and the destination are appraised, and the most

reliable ones are chosen for continued communication. By ignoring the tunnel path, the impact of periodic dropping and the Wormhole issue is reduced.

Sunil Kumar et al. [41] employed a friendship-based Jellyfish attack detection method to reduce Jellyfish attack activity in MANETs. A direct trust-based detection policy (DTD) is used to control packet collisions, congestion, and error spreading in the network. The results demonstrate that the malicious activities in the network are immediately and properly controlled.

For the removal of Jellyfish attacks in the network, the authors of [42] developed an authenticated routing system based on an attack detection framework that used a genetic fuzzy logic rule-based system. Fuzzification, rule evaluation, the aggregation of rule outputs, and defuzzification on the network are all examples of how fuzzy logic is applied in this study. This method is used with AODV as well as dynamic source routing (DSR) routing protocols and compared to other existing methods.

Doss Srinath et al. [20] proposed a new technique to ensure the prevention and detection of Jellyfish attacks in MANETs. This technique is a fusion of an authenticated routing-based framework for detecting Jellyfish attacks and a support vector machine (SVM) for learning packet forwarding behavior. The suggested technique uses the hierarchical trust evaluation property of nodes to select trusted nodes in a network for packet routing. The simulation results prove that the technique is quite effective in detecting Jellyfish attacks.

S. Satheeshkumar and N. Sengottaiyan [43] designed a clustering routing protocol on the basis of ant colony optimization. This protocol is a clustering of nodes that uses trust tables to determine a network's efficiency. This work produces beneficial results by increasing secure communication in a network and by detecting Jellyfish attacks in a highly effective manner.

Vamshi Krishna and Ganesh Reddy [44] implemented a delay-sensitive multipath selection algorithm to mitigate the Rushing attack in VANETs. The Rushing attack is an extreme and advanced sort of attack that can lead to Jellyfish or Byzantine attacks. In this work, they calculated each path delay by taking into account each vehicle's transmission, propagation, processing, and queuing delays, as well as geographical position. By checking the various delays, this approach successfully prevents Rushing attackers from entering the route discovery phase.

Amandeep Verma et al. [45] discussed security attacks and their related solutions. The authors created a multidimensional taxonomy of attacks by classifying them according to security services, attack layers, categories of attackers, and types of components targeted. After this, they compared relevant studies on the solution approaches to determine the advantages and disadvantages. Furthermore, Fonseca and Festag [46] studied and compared the security solutions that are currently in use based on performance standards and actual security measures. The authors also examined whether the properties of the chosen strategies satisfy VANETs' specifications.

According to numerous criteria, including detection strategy, attacks covered, simulators utilized, performance metrics, and limitations, in Table 2 we compare the security approaches outlined above. As we can observe, a variety of works have been proposed to deal with Jellyfish attacks and other attacks of a similar nature, such as DDoS, Blackholes, and Rushing. The pre-existing algorithms attempted to address the issue, but there is currently no known method of protecting against all three types of Jellyfish attacks. Therefore, our new protocol offers the successful prevention of various Jellyfish attack categories (Jellyfish reorder attacks, Jellyfish periodic drop attacks, and Jellyfish delay variance attacks) with a single solution.

**Table 2.** Comparison between detection techniques.

| Reference | Detection Strategy | Attack Covered | Network Type | Simulator | Performance Metrics | Approach Limitations |
|---|---|---|---|---|---|---|
| [32], 2013 | Hash function and time–space cryptography | Jellyfish reordering | Wireless ad hoc | Network simulator 2 (NS-2) | Congestion window, TCP goodput | The attacker node is not identified<br>Not compared with existing methods<br>More performance metrics are required |
| [38], 2014 | Genetic algorithm | Jellyfish periodic dropping | MANET | NS-2 | End-to-end delay, energy consumption, PDR, and throughput | The algorithm is not properly described<br>The accuracy of detection is not mentioned<br>Not compared with existing methods |
| [39], 2015 | Secure link establishment | Jellyfish reordering | MANET | ExataCyber | PDR, throughput, and end-to-end delay | The attacker node is not identified<br>Not compared with existing methods |
| [31], 2016 | Time measure | Jellyfish delay variance | MANET | NS-2 | Throughput, end-to-end delay | Not compared with existing methods<br>More performance metrics are required<br>The accuracy of detection is not mentioned |
| [37], 2017 | Intrusion detection system | Blackhole and Jellyfish | VANET | NS-2 | PDR, throughput, and end-to-end delay | Not compared with existing methods<br>The attacker node is not identified<br>Evaluated only for the AODV protocol<br>Only Jellyfish periodic dropping attacks are considered |
| [20], 2018 | Fusion of an authenticated routing-based framework and support vector machine | Jellyfish | MANET | NS-2 | Throughput, PDR, dropped packet ratio (DPR), and end-to-end delay | Not tested in the presence of attacks<br>The accuracy of detection is not discussed<br>Jellyfish reordering attacks are not considered |
| [41], 2018 | Direct trust-based detection | Jellyfish | MANET | MATLAB | Throughput, PDR, end-to-end delay, detection rate, false positive rate | The attacker node is not identified<br>Not compared with existing methods |

**Table 2.** *Cont.*

| | | | | | | |
|---|---|---|---|---|---|---|
| [26], 2018 | Trust scheme | Jellyfish delay variance | MANET | NS-2 | Throughput, end-to-end delay | The attacker node is not identified<br>Not compared with existing methods<br>More performance metrics are required |
| [34], 2019 | Trust-based evaluation | DDoS | VANET | NS-2 | PDR, latency, detection rate, and energy consumption | Computational overhead<br>Compared with only one other approach |
| [42], 2019 | Genetic fuzzy-based rule system | Jellyfish | MANET | NS-2 | Throughput, PDR, DPR, and end-to-end delay | Not tested in the presence of attacks<br>Computational overhead<br>Only Jellyfish periodic dropping attacks are considered |
| [35], 2020 | Trust-based intrusion detection system | DDoS | VANET | / | / | No simulation or evaluation |
| [36],2020 | Machine learning | DDoS | VANET | RStudio | Accuracy, minimum error rate | Invalidated parametric evaluation<br>Vehicle-to-vehicle (V2V) communication is not covered<br>High storage requirements |
| [44], 2021 | Delay sensitive multipath selection | Rushing | VANET | / | / | No simulation or evaluation |
| SecE-V2X | Trust-based | Jellyfish | V2X | Objective modular network testbed in C++ (OMNET++) | PDR, end-to-end delay, and throughput | Needs to be validated on a realistic dataset |

## 4. Secure and Efficient Routing Protocol for Jellyfish Attack Prevention in V2X (SecE-V2X)

Secure and efficient routing protocol for Jellyfish attack prevention in vehicle-to-everything (SecE-V2X) is a secured routing protocol based on the greedy perimeter stateless protocol (GPSR) by choosing the trusted next forwarding node from the current node's neighbors to defend the network against different Jellyfish attack categories.

Nodes acquire data utilized in the decision process when communicating with their neighbors' by recording the packet loss rate, delay, and packet reordering of each neighbor node, after which they then calculate their Honesty and store it locally in their table of neighbors. Table 3 is an example of a node neighbors' table.

**Table 3.** Table 3. Node neighbors' table in SecE-V2X.

| Node-Id | Position Information (x,y) | Overall-Honesty | Timestamp |
|---------|---------------------------|-----------------|-----------|
| 1 | 6218.79, 2363.39 | 0.75 | 22.75 |
| 2 | 6552.16, 2286.45 | 0.63 | 23.87 |
| 3 | 6536.72, 2181.74 | 0.86 | 23.42 |

In addition to the GPSR beacon that is broadcasted periodically, the proposed protocol broadcasts an Honesty beacon message containing the updated information after any neighbor node Honesty update. Figure 4 depicts the Honesty update beacon structure in our proposed strategy.

| Node-Id | Position-Information | Neighbor-Id | Neighbor-Honesty | Timestamp |
|---------|----------------------|-------------|------------------|-----------|

**Figure 4.** The Honesty beacon format in SecE-V2X.

When a node decides to send data to a destination, it first chooses the optimal forwarding node with the best Honesty value based on information stored in its neighbors' table and then sends the data to that node. The chosen node then transmits the received data to the next node in the same way, and the procedure is continued until the destination is reached. In this manner, the malicious nodes will be isolated from participating in the routing process. The selected relay node is shown in Figure 5 based on the proposed next-hop selection method. Despite not being the closest to the destination (D) in this instance, the source (S) chooses node 3. As seen in Table, compared to other neighbors of (S), node 3 has a higher value of Honesty.

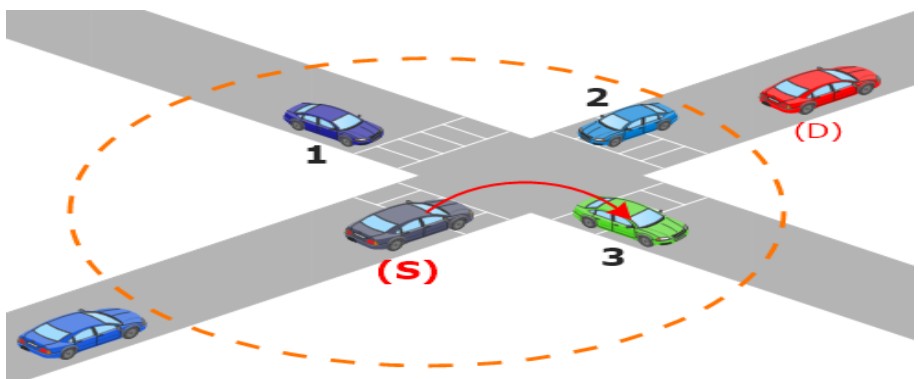

**Figure 5.** Choosing the next hop in the SecE-V2X protocol.

The overall Honesty value of a node is calculated by the communication behavior of packet delivery between nodes which is represented in the packet loss rate, delay, packet reordering, and confidence of the node. The Honesty value is set in a range from 0 to 1,

where 0 is completely dishonest and 1 is completely honest. Because the network has a dynamic topology and is self-organizing, nodes will join and leave the network at random. As a result, node Honesty must be updated in real time. This value changes at any given time according to the transactions between nodes. An overview of the entire work of the proposed protocol is presented in Figure 6. The presentation of the Honesty variables as well as the technique of next-hop selection employed in the proposed protocol take up the remainder of this section.

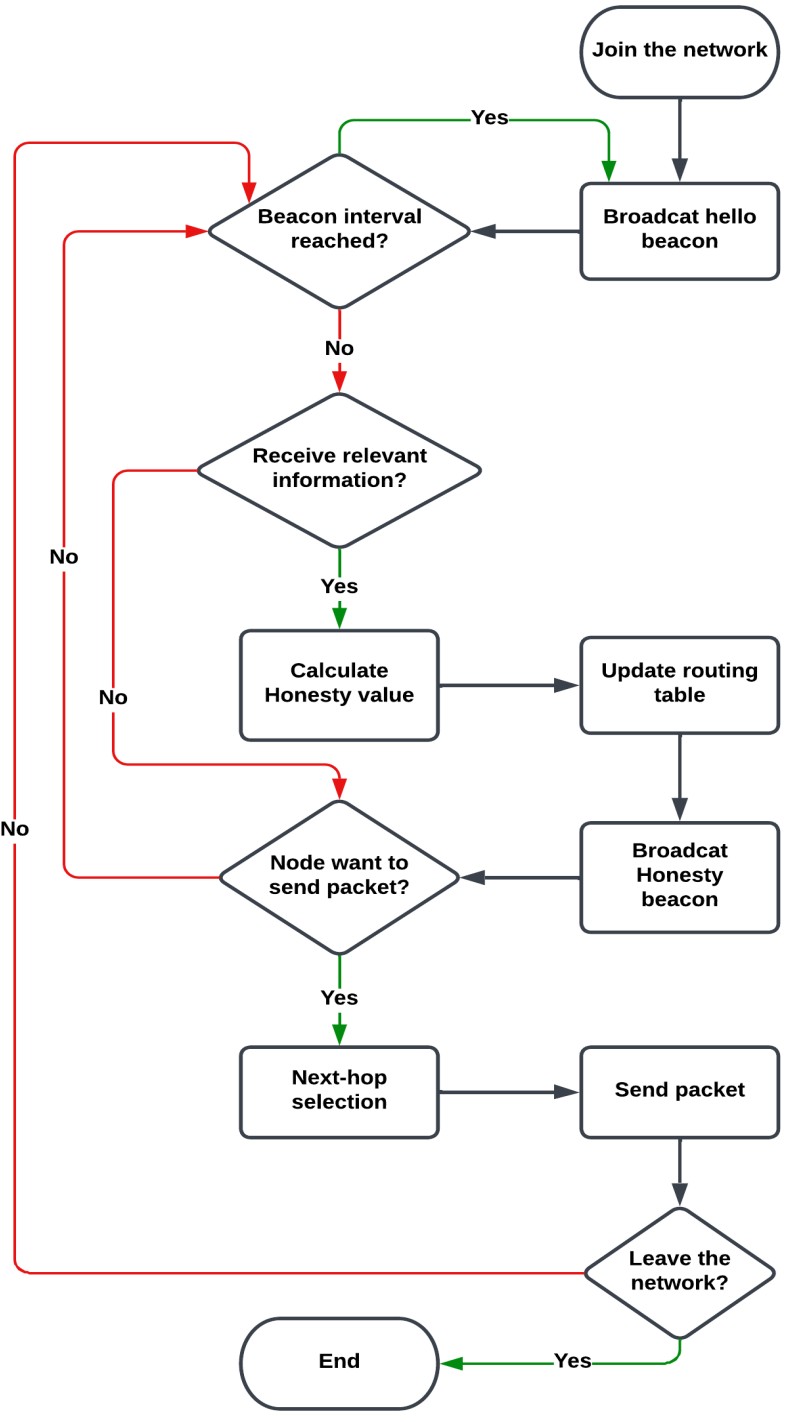

**Figure 6.** The flow chart of the entire working of the proposed protocol.

As a result, SecE-V2X's message complexity is equal to N*M, where N is the number of nodes, since each needs to initially broadcast a beacon, and M is the number of neighbors, since each vehicle calculates the Honesty value and sends this value to its neighbors. Additionally, because a node must wait for $\varrho$ time units before broadcasting the beacon again, the algorithm's time complexity is equal to the timer duration, $\varrho$.

*4.1. Honesty Characteristics*

In designing our SecE-V2X protocol, we consider the following characteristics:

- Only honest nodes can participate in the routing path.
- Source and destination nodes are considered honest nodes.
- Lack of symmetry: This can be defined as the lack of Honesty equivalence between nodes. If node U is honest to node V, it does not necessarily mean that node V is honest to node U.
- Transitive relation: The honest relationship between two nodes can be provided to other nodes as recommended confidence.
- Composite: The honest values collected from many possible paths can be combined to produce an integrated honest value.

*4.2. Honesty Metrics*

The main impact of Jellyfish attacks on a network is that they have a significant influence on throughput and end-to-end delay. Regarding the effects of these attacks, we choose the appropriate metric for every Jellyfish category. In the proposed protocol, three Honesty metrics are used: packet delay, packet loss rate, and packet reordering. By computing these metrics, our proposed protocol mitigates the Jellyfish nodes from participating in the routing process. In the subsections that follow, these metrics are described in detail.

4.2.1. Packet Delay

The delay rate of a node is essentially stable after sending or receiving data in a network with an invariable environment, or it swings with a certain tolerance; however, with Jellyfish nodes, the delay rate is much higher than the rate of normal nodes. Delay can be defined as the average time incurred by transmitting data to neighboring nodes. It can be calculated using Equation (1), proposed in [47]:

$$D = t_r - t_s \tag{1}$$

where $t_r$ and $t_s$ are the reception and sending time of a packet, respectively. This metric is used to discover Jellyfish delay variance attacks.

4.2.2. Packet Loss Rate

The packet loss rate refers to the ratio of lost packets to the total packets transmitted or received over a given time period. In order to handle Jellyfish periodic dropping attacks, Equation (2), introduced in [48], is used to calculate the packet loss rate, $LR(n)$, of a node, $n$:

$$LR(n) = \frac{PRD + PSD}{PRC + PSC + PRD} \tag{2}$$

where *PSC* is the number of packets sent correctly, *PRC* is the number of packets received correctly, *PSD* is the number of packets sent but dropped, and *PRD* is the number of packets received but dropped.

### 4.2.3. Packet Reordering

To deal with Jellyfish reordering attacks and capture a packet's displacements from their original locations, this metric is defined as the percentage of packets that are late relative to their predicted position, as determined by the received index. The packet reordering, *PR*, is calculated using Equation (3), proposed in [49]:

$$
\begin{cases}
PR = \sum\limits_{i=+1}^{i=D_r} RD[i] \\
D[j] = RI[j] - AS[j]
\end{cases}
\tag{3}
$$

where *D[j]* is the displacement of the packet, *j*, *RI[j]* is the received index of the packet, *j*, and *AS[j]* is the arrived sequence of the packet, *j*. Reordering density (*RD*), proposed in [50], refers to the distribution of packet displacements from their sending positions. As a result, *PR* = 0 corresponds to the case that all of the packets are in the correct order and *PR* > 0 for the reordered packet sequence.

### 4.3. Calculating the Node Honesty

To obtain a neighbor node's Honesty, a node computes the direct and the recommended confidence. In this subsection, we present how to calculate the node Honesty in the proposed protocol.

### 4.3.1. Communication Confidence

The communication confidence can be calculated based on the number of transmitted packets. Communication confidence is calculated using Equation (4), presented in [51]:

$$
\begin{cases}
C_{com} = \frac{2C_{frd} + C_{fct}}{2} \\
C_{frd} = \frac{2s_p + 1}{2\gamma} \\
C_{fct} = \frac{1}{\gamma} \\
\gamma = s_p + f_p + 1
\end{cases}
\tag{4}
$$

where $s_p$ is the number of successfully delivered packets, $f_p$ is the number of unsuccessfully delivered packets, $C_{com}$ is communication confidence, $C_{frd}$ is forward packet confidence, and $C_{fct}$ is factors of uncertainty confidence.

### 4.3.2. Direct Confidence

The direct confidence can be calculated by integrating the communication confidence, delay, and packet reordering, *PR*, using Equation (5):

$$
C_{direct} = \begin{cases}
w_{com} \times C_{com} + \frac{w_{Delay}}{D} & PR = 0 \\
0 & PR > 0
\end{cases}
\tag{5}
$$

$C_{com}$, *D*, and *PR* are the communication confidence from Equation (4), delay from Equation (1) and packet reordering calculated in Equation (3), respectively. $w_{com}$ and $w_{Delay}$ are the weight adjustment parameters of communication confidence and delay, respectively, with $w_{com} + w_{Delay} = 1$ and $w_{com}, w_{Delay} \in [0, 1]$.

### 4.3.3. Recommended Confidence

The calculation of confidence includes both direct and recommended confidence. The recommended confidence is calculated based on the indirect information provided by the neighboring nodes' recommendations through the use of Equation (6):

$$
C_{rcd} = \frac{\sum_{i=1}^{n} H_{ovr}(i)}{n}
\tag{6}
$$

where $n$ is the number of neighbors and $H_{ovr}$ *(i)* is the Honesty recommended from a neighbor, *i* (Equation (7)).

### 4.3.4. Overall Honesty

The integrated Honesty ($H_{ovr}$) is calculated by direct as well as recommended confidence dependent upon the individual effect of that kind of confidence. To address the nodes that provide misleading recommended confidence, we used *α* and *β* weights to underestimate the effect of false confidence. The overall Honesty is obtained using Equation (7), where *α*, *β* ∈ [0, 1], *α* + *β* = 1, and *α* > *β*:

$$H_{ovr} = \alpha \times C_{direct} + \beta \times C_{rcd} \tag{7}$$

### 4.4. SecE-V2X Routing Algorithm

The SecE-V2X algorithm ensures security in vehicular networks by isolating malicious nodes from participating in packet routing during the next-hop selection phase. To ensure this, when a node communicates with any neighbor the Honesty value of this neighbor is computed in Equation (7) through the use of the packet loss rate, delay, and packet reordering. The beacon containing the updated Honesty is broadcasted after every Honesty update. Algoritham 1 below describes the steps involved to calculate the neighbor's Honesty value.

---

**Algorithm 1.** Steps involved when calculating the Honesty value for the neighbor (N).

---

```
// The steps taken to calculate the neighbor Honesty value
With the neighbor N do
D←getDelay(N); // Getting the delay of the node N
Sp←getSp(N);     // Getting the number of successfully delivered packets of the node N
Fp←getFp(N);     // Getting the number of unsuccessfully delivered packets of the node N
y←Sp+Fp+1;
Cfrd←(2*Sp +1)/(2*y); // Calculating the forward packet confidence
Cfct←1/y; // Calculating the factor of uncertainty confidence
Ccom← (2*Cfrd+Cfct)/2       // Calculating the communication confidence
PR←getPR(N);     // Getting the packet reordering value of the node N
if PR=0 then
Cdirect← Wcom*Ccom+Wdelay/D        // Calculating the direct confidence
else
Cdirect←0;
end if
Crcd←0;
for all i in neighborsTable do
      Crcd←Crcd+getHovr(N)[i];        // Calculating the sum of Recommended confidences
end for
Crcd←Crcd/neighborsTable.length;        // Calculating the average of the Recommended confidences
Hovr←a*Cdirect+b*Crcd; // Calculating the overall honesty
My-id ← getNodeId(); //Getting current node id
Neighbor-id ← getNodeId(N); //Getting neighbor Node id
MyPosition← getPosition(); // Getting current node position
NPosition← getPosition(N); // Getting neighbor N position
t ← now(); //Getting current time
neighborsTable.upDateHonesty(Neighbor-id, NPosition, Hovr, t); //Updating neighborsTable
sendHonestyBeacon (My-Id, MyPosition, Neighbor-id, Hovr, t);
```

---

The recommended confidence and the Honesty value are computed after receiving an Honesty beacon message, and the table of neighbors is then updated. Algorithm 2 displays the activities conducted when receiving an Honesty beacon message.

---

**Algorithm 2.** Steps required when receiving an Honesty beacon message.

---

// The steps taken when receiving an Honesty beacon message
**extractHonestyBeacon**(Id, Position, Neighbor-id, Neighbor-Honesty, Timestamp);        //Extracting information from the beacon message
setHovr(N)[Id]←Neighbor-Honesty;        // Setting the honesty of the node N provided from the node Id
Crcd←0;
**for** all i in neighborsTable **do**
        Crcd←Crcd+getHovr(N)[i]; // Calculating the Recommended confidence by getting the honesty of the node N provided from the node i
**end for**
Hovr←a*Cdirect+b*Crcd;        // Calculating the overall honesty
neighborsTable.**upDateHonesty**(Neighbor-id, Neighbor-Position, Hovr, now()); //Updating node information in neighborsTable

---

When a node intends to transmit data to another node, it first examines all of its neighbors, after which it then chooses the one with the highest Honesty value and delivers the packet to that node. Until it arrives at its destination, the intended receiver follows the same procedure. The next forwarding nodes from the source to the destination are picked as shown in Figure 7.

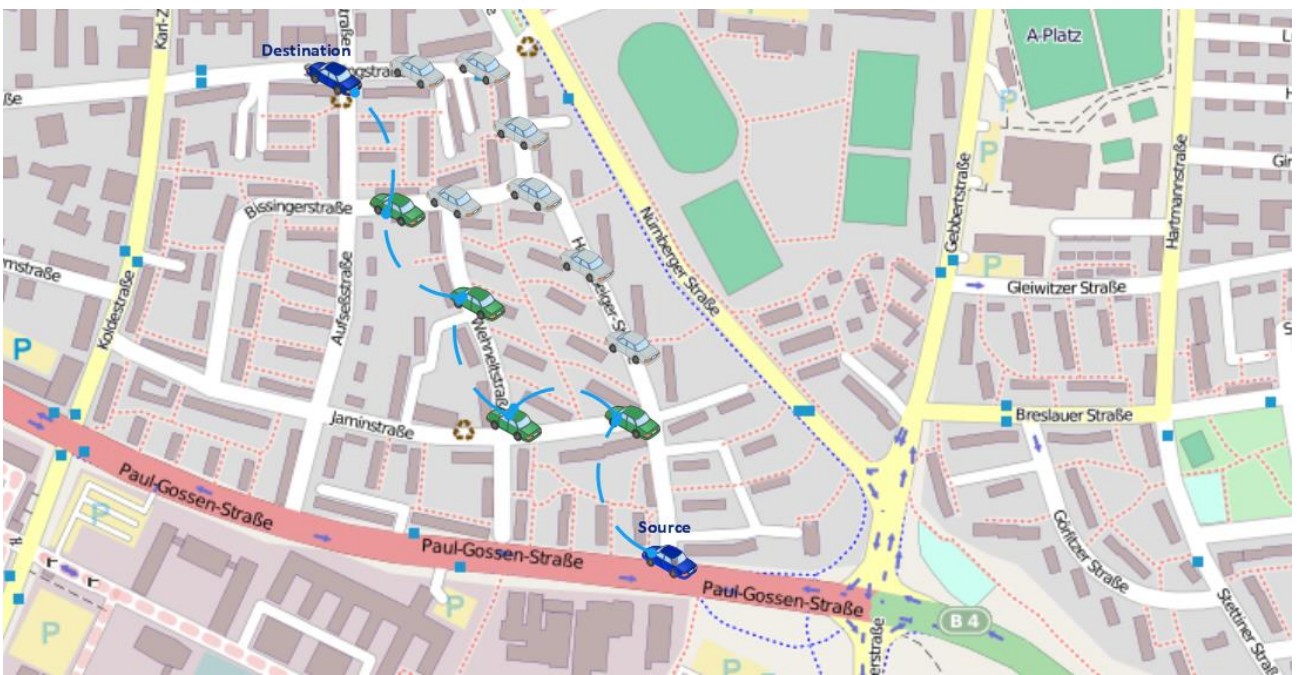

**Figure 7.** Next-hop selection from source to destination.

When all of the neighbors' Honesty values fall below the desired weight level, the current node sends the identical packet to the two best neighbors at the same time to increase the chances of receiving the message correctly. Algorithm 3 describes the SecE-V2X next-hop selection procedure.



---

**Algorithm 3.** The next-hop selection steps in the SecE-V2X protocol.

---

```
// The steps taken while deciding to deliver a data packet
selfPosition ← getPosition();        // Getting current node position
destinationId ← getNodeId(destination);         // Getting the ID of the destination node
destinationPosition ← getPosition(destination);       // Getting the position of the destination node
myDistance ← (destinationPosition–selfPosition).length();        //Calculate the distance between the current node and the
destination
if destinationId in neighborsTable then         //Check if the destination in neighborsTable
      sendPacketTo(destinationId); //Sending the packet to the destination
else
      bestHonesty ← 0;
      for all i in neighborsTable do
           neighborDistance ← (destinationPosition–neighborPosition).length();        // Calculate the distance between the
neighbor node and the destination
           if (myDistance > neighborDistance) then
                if (neighborHonesty > bestHonesty) then // Choosing the neighbor with the best honesty
                     bestHonesty ← neighborHonesty;
                     bestNeighbor ← getNodeId(neighbor);
                end if
           end if
      end for
      if (bestHonesty > weightLevel) then       //Check the weight level
           sendPacketTo (bestNeighbor); // Transferring the packet to the chosen next-hop node
      else
           chooseTwoBestNeighbors(bestNeighbor1, bestNeighbor2); // choosing the best-honesty pair of nodes
           sendPacketTo(bestNeighbor1);
           sendPacketTo(bestNeighbor2);
      end if
end if
```

---

## 5. Result Analysis and Performance Evaluation

In this section, we present the tools, parameters, and metrics used to create the network scenario and validate our proposed SecE-V2X protocol for defending against Jellyfish attacks. The simulator for urban mobility (SUMO) [52] was employed to generate nodes' mobility traffic urban scenarios, as shown in Figure 8. INET [53] and vehicles in network simulation (Veins) [54] frameworks were developed using the objective modular network testbed in C++ (OMNET++) network simulator [55], and are used for communication between nodes.

The simulation scenario used is an urban environment of 2500 m × 2500 m in Erlangen. The speeds of the nodes vary from 0 to 14 m/s. We perform each simulation both under and without attacks for 500 s with a node density of 100 nodes. The main parameters of our simulation are summarized in Table 4.

The proposed protocol was tested against all three Jellyfish attack types for various numbers of malicious nodes (0, 3, 6, and 9). The malicious nodes are placed in the middle of node pools without any collusion. Figure 9 shows a real-time simulation scenario with a simulation time of 200 s, where malicious nodes are identified in red. We compare the performance of our proposed protocol with two secured routing approaches from [37], 2017, and [20], 2018, while considering the effects of malicious nodes on the network. We re-simulated the algorithms in the mentioned approaches in order to ensure a correct and fair comparison in terms of tools, network density, mobility models, node speed, etc. As Jellyfish attacks degrade network performance and reduce all flows' throughputs by dropping and delaying packets, through this experimental study we intend to evaluate the efficiency of SecE-V2X in terms of optimizing nodes' packet delivery ratios as well as data throughputs, in addition to reducing the transmission delay.

### 5.1. Under Jellyfish Periodic Drop Attacks

The function of Jellyfish periodic drop attacks in our simulations is to drop the packets periodically for 3 s every 7 s. In this section, our primary concern is evaluating the effectiveness of our recently developed protocol in the presence of Jellyfish periodic drop attacks. The performance comparison values are given in Table 5.

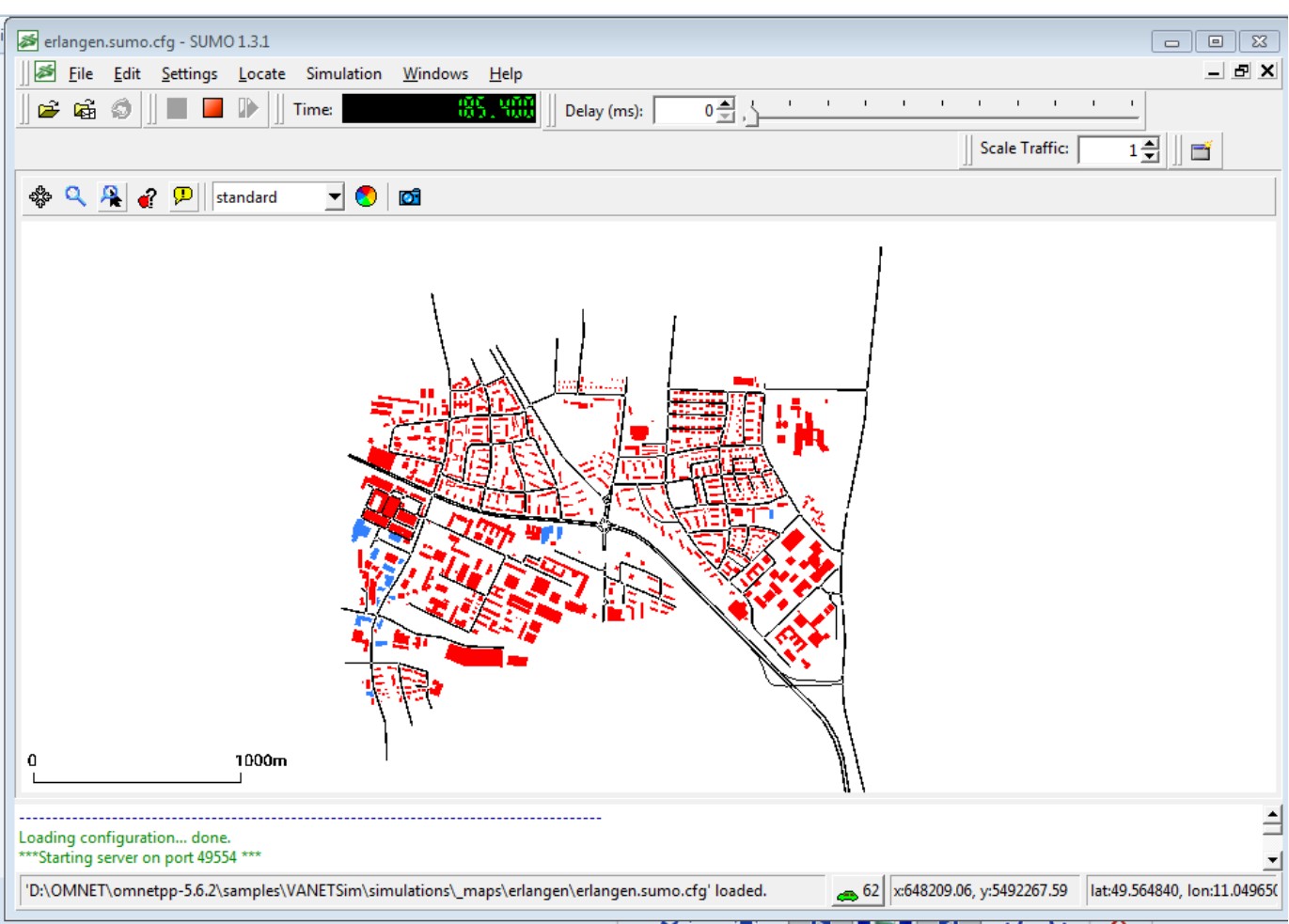

**Figure 8.** Mobility traffic scenario.

**Table 4.** Simulation parameters.

| Parameter | Value |
|---|---|
| OMNET++ version | 5.2.6 |
| INET version | 4.2.2 |
| Veins version | 5.0 |
| Environment | Urban |
| SUMO version | 1.3.1 |
| Simulation area | 2500 m × 2500 m |
| Simulation time | 500 s |
| Number of nodes | 100 |
| Number of attackers | 0, 3, 6, and 9 |
| Max node speed | 14 m/s |
| Mobility model | Erlangen |
| MAC protocol | IEEE 802.11p |
| Transmission range | 250 m |

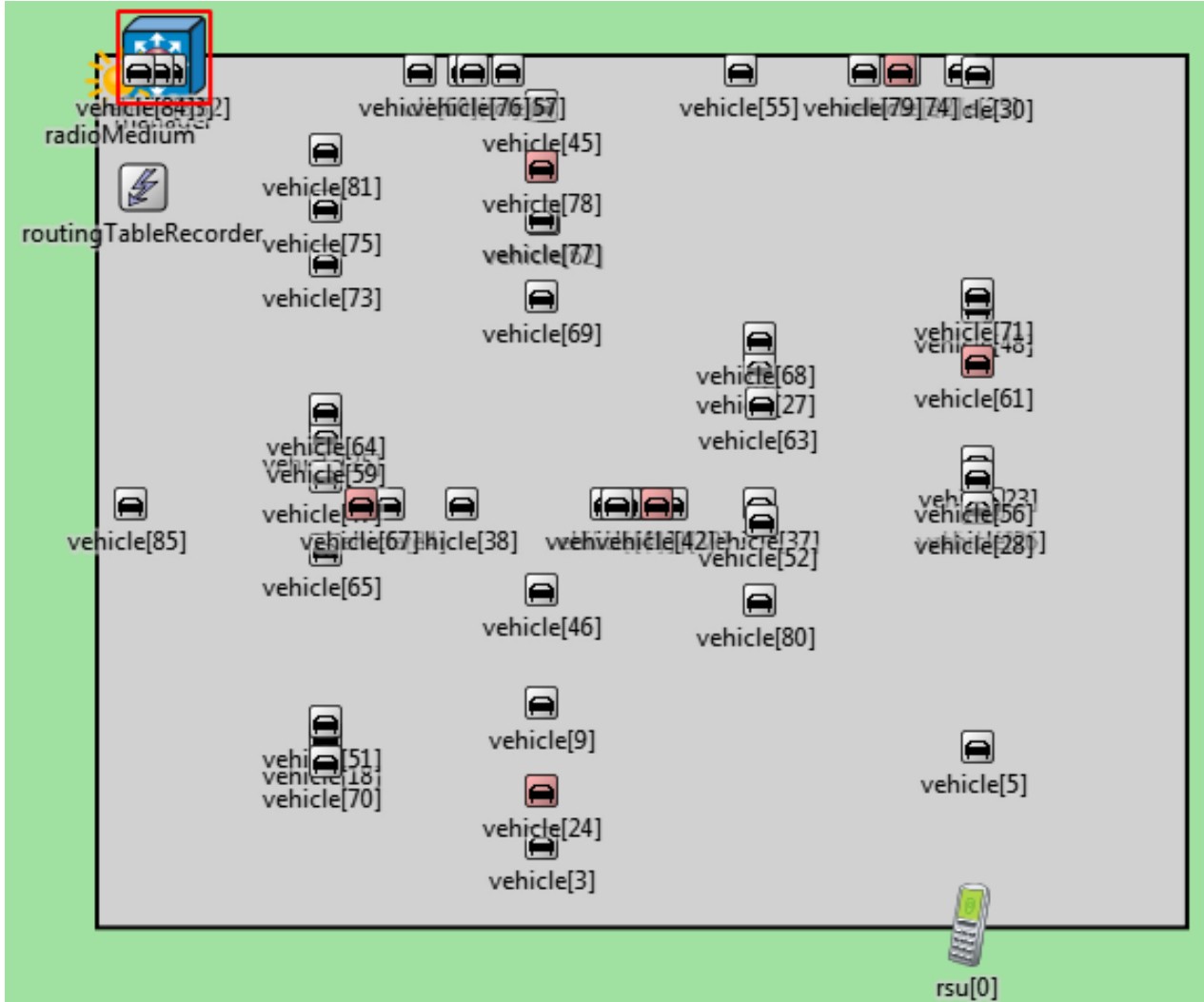

**Figure 9.** Simulation scenario.

**Table 5.** Performance values comparison under Jellyfish periodic drop attacks.

| Number of Attackers | Metrics | SecE-V2X | [20], 2018 | [37], 2017 |
|---|---|---|---|---|
| 0 | PDR (%) | 95.34 | 93.80 | 91.60 |
| | EED (ms) | 6.80 | 7.11 | 8.64 |
| | Throughput (packets/s) | 638.15 | 596.33 | 534.55 |
| 3 | PDR (%) | 93.21 | 88.76 | 79.63 |
| | EED (ms) | 7.48 | 7.94 | 12.21 |
| | Throughput (packets/s) | 608.22 | 556.65 | 452.11 |
| 6 | PDR (%) | 89.67 | 76.72 | 64.67 |
| | EED (ms) | 8.31 | 9.64 | 16.83 |
| | Throughput (packets/s) | 551.29 | 460.54 | 346.86 |
| 9 | PDR (%) | 76.83 | 69.75 | 55.29 |
| | EED (ms) | 10.48 | 13.29 | 21.34 |
| | Throughput (packets/s) | 433.53 | 398.41 | 282.38 |

5.1.1. Packet Delivery Ratio (PDR)

The packet delivery ratio can be calculated as the proportion of the total number of packets successfully received at the destination node as compared to the number of packets sent by the source node. This metric is measured as follows:

$$\text{PDR} = \frac{\Sigma \text{Packets}_{\text{received}}}{\Sigma \text{Packets}_{\text{sent}}} * 100 \tag{8}$$

where $\text{Packets}_{\text{received}}$ is the successfully received packets by the destination node and $\text{Packets}_{\text{sent}}$ is all of the packets sent by the source node.

In accordance with the number of attacker nodes, Figure 10 compares the variations in the PDR of the proposed SecE-V2X with the aforementioned security algorithms. For all of the protocols, it can be observed that the PDR decreases whenever the number of malicious nodes in the network grows. In a network without any malicious nodes, we note that the suggested protocol produces better results than those of the other schemes and has a higher PDR of almost 95.5%. The outcomes for three malicious nodes indicate that SecE-V2X performs at 93.21%, significantly better in terms of the PDR than [37], 2017 (74.21%), and even [20], 2018 (90.22%), demonstrating that our recently designed protocol works effectively even in the presence of a few malicious nodes. When there are six malicious nodes, the proposed protocol's PDR drops somewhat to 89.67%, which is still much higher than [37], 2017 (about 64.67%), and [20], 2018 (about 76.72%). The PDR of our suggested protocol reduces significantly to 76.83% when nine malicious nodes are taken into account, since the number of attacker nodes is exaggerated, which degrades the network. The results show that the Honesty metrics of SecE-V2X allow it to use the reading of the networks to make a decision about the best route to take in terms of the PDR.

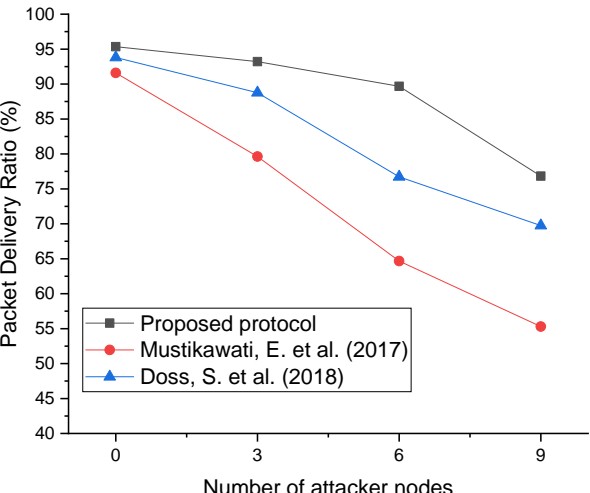

**Figure 10.** Packet delivery ratio under Jellyfish periodic drop attacks [20,37].

5.1.2. End-to-End Delay

The end-to-end delay reflects the average amount of time that data packets take to travel from their source to their destination node. As shown in Equation (9), it can be calculated by dividing the sum of the times for all received packets by the total number of packets:

$$\text{EEDelay} = \frac{\Sigma (\text{Packet}_{\text{received time}} - \text{Packet}_{\text{sent time}})}{\text{Nbr}_{\text{Packets}}} \tag{9}$$

where $\text{Packet}_{\text{received time}}$ is the packet received time, $\text{Packet}_{\text{sent time}}$ is the packet sending time, and $\text{Nbr}_{\text{Packets}}$ is the number of delivered packets.

Figure 11 shows the end-to-end delay performance comparison results of the SecE-V2X algorithm with the aforementioned security algorithms. As more malicious nodes

are deployed, the end-to-end delay value tends to decrease. According to the results, the proposed SecE-V2X has a lower end-to-end delay value than that of the existing algorithms for any number of attacker nodes. When there are no malicious nodes, we observe that the suggested protocol outperforms the competing approaches and has a reduced end-to-end delay value of almost 6.8 ms. In the presence of three malicious nodes, the results show that the end-to-end delay value of SecE-V2X is 7.48 ms, better in terms of end-to-end delay than [37], 2017 (12.21 ms), and even [20], 2018 (7.94 ms). When the number of attacker nodes increases to six, the end-to-end delay of the proposed protocol increases slightly to 8.31 ms, which is still significantly better than [37], 2017 (16.83 ms) and [20], 2018 (9.64 ms). With nine malicious nodes, the end-to-end delay degrades to 10.48 ms for the proposed protocol, 21.34 ms for [37], 2017, and 13.29 ms for [20], 2018. This can be explained by the fact that the packet delay metric allows our SecE-V2X to favor the selected routes with better end-to-end delays.

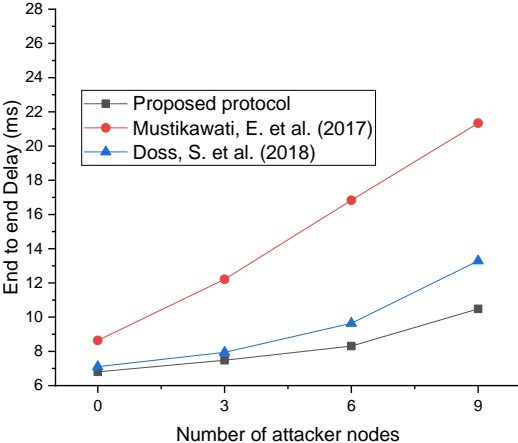

**Figure 11.** End-to-end delay under Jellyfish periodic drop attacks [20,37].

### 5.1.3. Throughput

The throughput is the number of delivered data packets within a given time frame. It can be calculated as follows:

$$\text{Throughput} = \frac{\text{Nbr}_{\text{Packets}}}{\text{T}_{\text{Period}}} \tag{10}$$

where $\text{Nbr}_{\text{Packets}}$ is the number of delivered packets and $\text{T}_{\text{Period}}$ is a particular time period.

An ideal throughput is produced by a high PDR and a low end-to-end delay. Figure 12 shows the proposed SecE-V2X's generated throughput depending on the number of attacker nodes. We can observe, without a doubt, that our proposed protocol maintains better throughput than that of the competing schemes. In the absence of attacker nodes, the proposed protocol has an average throughput of 638.15 packets/s, while [37], 2017, had a throughput of 534.55 packets/s and [20], 2018, had a throughput of 596.33. The average throughput of the proposed protocol in the presence of three malicious nodes is 608.22 packet/s, whereas [37], 2017, has a throughput of 452.11 packet/s and the throughput of [20], 2018, is 556.65 packet/s. We can clearly see that the throughput performance drops drastically with an increase in the number of attacker nodes in the network. The throughput goes down to the lowest with nine malicious nodes, reaching 433.53 packets/s in the proposed protocol, 282.38 packets/s in [37], 2017, and 398.41 packets/s in [20], 2018. Through the Honesty metrics, our protocol can obtain better performance with regard to throughput.

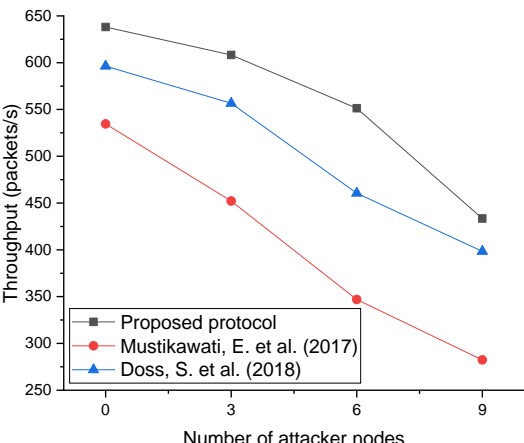

**Figure 12.** Nodes' data throughput under Jellyfish periodic drop attacks [20,37].

## 5.2. Under Jellyfish Delay Variance Attacks

In our simulations, Jellyfish delay variance attacks introduce a random delay between 100 and 300 milliseconds before transmitting packets. In this section, we intend to evaluate the efficiency of SecE-V2X in the presence of Jellyfish delay variance attacks. Table 6 provides the performance comparison values.

**Table 6.** Performance values comparison under Jellyfish delay variance attacks.

| Number of Attackers | Metrics | SecE-V2X | [20], 2018 | [37], 2017 |
|---|---|---|---|---|
| 0 | PDR (%) | 95.34 | 93.80 | 91.60 |
| | EED (ms) | 6.80 | 7.11 | 8.64 |
| | Throughput (packets/s) | 638.15 | 596.33 | 534.55 |
| 3 | PDR (%) | 94.62 | 90.38 | 80.20 |
| | EED (ms) | 7.92 | 9.34 | 13.84 |
| | Throughput (packets/s) | 617.42 | 562.19 | 455.41 |
| 6 | PDR (%) | 91.33 | 81.65 | 69.47 |
| | EED (ms) | 9.11 | 13.33 | 17.59 |
| | Throughput (packets/s) | 560.75 | 483.62 | 383.34 |
| 9 | PDR (%) | 79.53 | 71.93 | 58.42 |
| | EED (ms) | 11.28 | 15.09 | 23.14 |
| | Throughput (packets/s) | 448.76 | 400.86 | 298.36 |

### 5.2.1. Packet Delivery Ratio (PDR)

An illustration of how delay variance attacks affect the PDR is provided in Figure 13. Similar to periodic drop attacks, with slightly higher values in delay variance attacks the PDR decreases whenever the number of malicious nodes grows. For three malicious nodes, the proposed protocol performs at 94.62%, higher than [37], 2017 (about 80.2%), and [20], 2018 (about 90.38%). When the number of attacker nodes is exaggerated to reach nine, the PDR of [37], 2017, reduces significantly to 58.42%, while [20], 2018, has a PDR of 71.93%, and our proposed protocol maintains the best PDR of 79.53%. The outcomes indicate that SecE-V2X performs significantly better in terms of the PDR than [37], 2017, and [20], 2018, demonstrating that our recently designed protocol works effectively even in the presence of Jellyfish delay variance attacks.

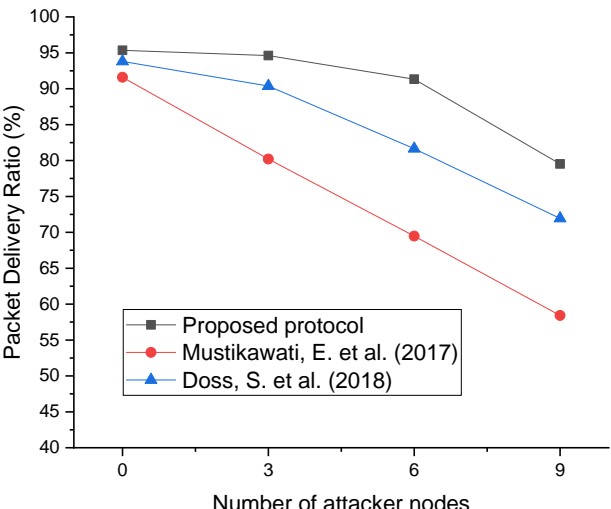

**Figure 13.** Packet delivery ratio under Jellyfish delay variance attacks [20,37].

### 5.2.2. End-to-End Delay

The end-to-end delay ratio for different numbers of delay variance attacker nodes is demonstrated in Figure 14. Since this type of attack introduces some delay before transmitting packets, the end-to-end delay values are slightly higher than in periodic drop attacks. With a ratio of 7.92 ms, the proposed protocol with the lowest delay outperforms [37], 2017 (13.84 ms), and [20], 2018 (9.34 ms), for three malicious nodes. The end-to-end delay of [37], 2017, significantly increases to 23.14 ms when the number of attacker nodes is inflated to nine, whereas [20], 2018, has an end-to-end delay of 15.09 m; our proposed protocol maintains the best end-to-end delay of 11.28 ms. The results show that SecE-V2X outperforms [37], 2017, and [20], 2018, in terms of end-to-end delay, proving that our developed protocol is still successful in the face of Jellyfish delay variance attacks.

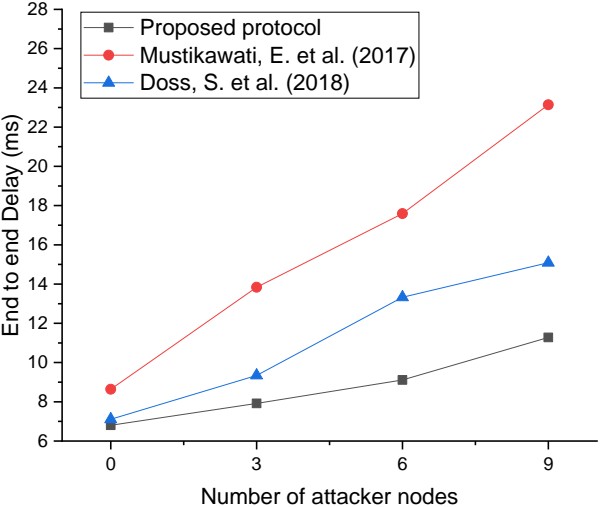

**Figure 14.** End-to-end delay under Jellyfish delay variance attacks [20,37].

### 5.2.3. Throughput

Figure 15 shows a comparison of throughput depending on various numbers of delay variance attacker nodes. In delay variance attacks, the throughput decreases whenever the number of attackers increases. The results show that, for any number of attacker nodes, the proposed SecE-V2X has a higher throughput value than that of the other algorithms. For three malicious nodes, the proposed protocol performs 617.42 packets/s, better than [37], 2017 (455.41 packets/s), and [20], 2018 (562.19 packets/s). The throughput of [37], 2017,

drastically reduces to 298.36 packets/s when the number of attacker nodes is increased to nine, whereas [20], 2018, has a throughput of 400.86 packets/s, and our protocol retains the best throughput of 448.76 packets/s. The results demonstrated that SecE-V2X performs better than [37], 2017, and [20], 2018, in terms of throughput, demonstrating that our proposed protocol remains effective against Jellyfish delay variance attacks.

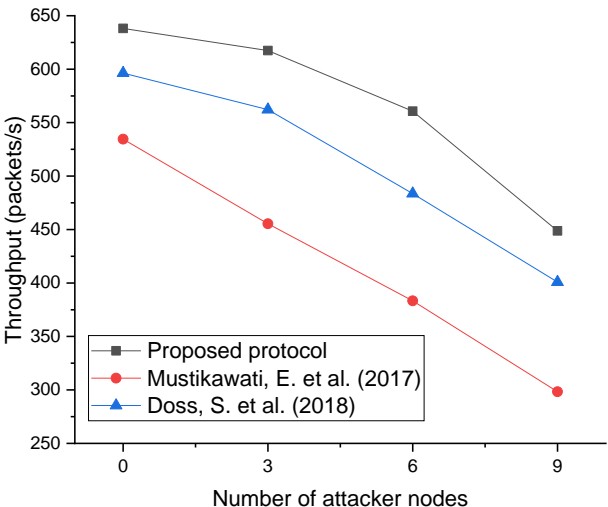

**Figure 15.** Nodes' data throughput under Jellyfish delay variance attacks [20,37].

### 5.3. Under Jellyfish Reordering Attacks

In Jellyfish reordering attacks, the attacker node selects packets at random before transmitting them. Our main focus in this section is evaluating and comparing the performance of our protocol in the presence of this attack. The performance comparison values are provided in Table 7.

**Table 7.** Performance values comparison under Jellyfish reordering attacks.

| Number of Attackers | Metrics | SecE-V2X | [20], 2018 | [37], 2017 |
|---|---|---|---|---|
| 0 | PDR (%) | 95.34 | 93.80 | 91.60 |
| | EED (ms) | 6.80 | 7.11 | 8.64 |
| | Throughput (packets/s) | 638.15 | 596.33 | 534.55 |
| 3 | PDR (%) | 93.92 | 89.57 | 79.92 |
| | EED (ms) | 7.70 | 8.64 | 13.03 |
| | Throughput (packets/s) | 612.82 | 559.42 | 453.76 |
| 6 | PDR (%) | 90.50 | 79.19 | 72.07 |
| | EED (ms) | 8.71 | 11.49 | 17.21 |
| | Throughput (packets/s) | 556.02 | 472.08 | 402.10 |
| 9 | PDR (%) | 78.18 | 74.84 | 56.86 |
| | EED (ms) | 10.88 | 14.19 | 22.24 |
| | Throughput (packets/s) | 441.15 | 415.64 | 290.37 |

#### 5.3.1. Packet Delivery Ratio (PDR)

The impact of Jellyfish reordering attacks on the PDR is depicted in Figure 16. Similar to periodic drop and delay variance attacks, the PDR decreases whenever the number of attackers increases. In comparison to [37], 2017 (about 79.91%), and [20], 2018 (about 89.57%), the proposed protocol performs at a higher ratio of 93.91% for three malicious nodes. The PDR of [37], 2017, drastically drops to 56.85% when the number of attacker nodes is raised to nine, whereas [20], 2018, has a PDR of 74.84%, and our proposed protocol has the highest PDR of 78.18%. The results show that SecE-V2X performs noticeably better

in terms of the PDR than other approaches, indicating that our protocol performs efficiently in the presence of Jellyfish reordering attacks.

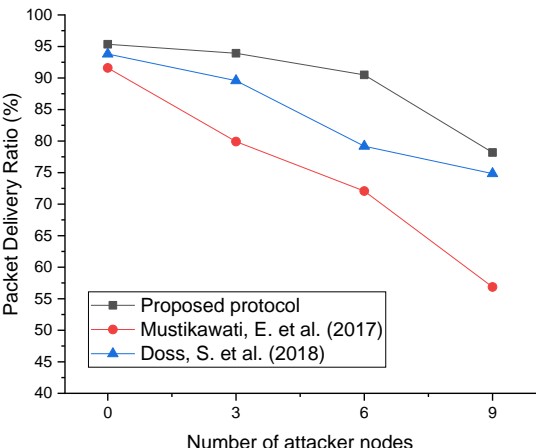

**Figure 16.** Packet delivery ratio under Jellyfish reordering attacks [20,37].

### 5.3.2. End-to-End Delay

Figure 17 illustrates the end-to-end delay ratio for various reordering attacker node numbers. For three malicious nodes, the proposed protocol, with the shortest delay of 7.7 ms, performs better than [37], 2017 (13.02 ms), and [20], 2018 (8.64 ms). When the number of malicious nodes is raised to nine, the end-to-end delay of [37], 2017, considerably increases to 22.24 ms. In contrast, [20], 2018, has an end-to-end delay of 14.19 ms, while our proposed protocol has the best end-to-end delay of 10.88 ms. According to the results, the proposed protocol exceeds the competing approaches, demonstrating that our proposed protocol is effective against Jellyfish reordering attacks.

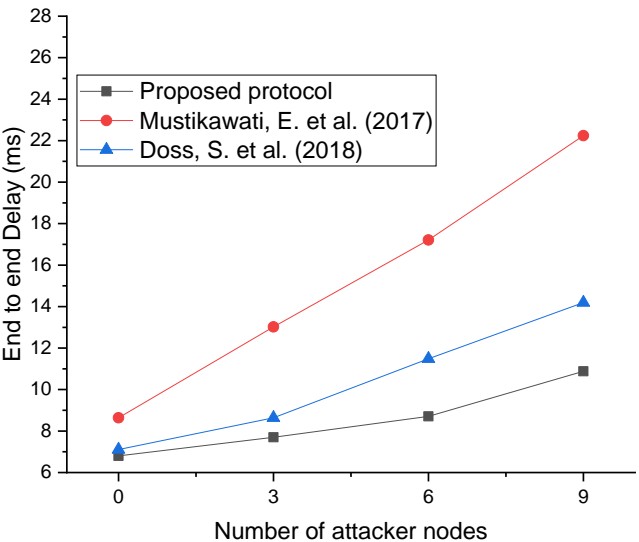

**Figure 17.** End-to-end delay under Jellyfish reordering attacks [20,37].

### 5.3.3. Throughput

A comparison of performance based on different numbers of reordering attack nodes is demonstrated in Figure 18. The throughput in Jellyfish reordering attacks drops as the number of malicious nodes increases. The outcomes indicate that SecE-V2X has a greater throughput value than that of other algorithms for different numbers of malicious nodes. Our protocol outperforms [37], 2017 (453.76 packets/s), and [20], 2018 (559.42 packets/s), for three malicious nodes by 612.82 packets per second. When there are

nine attacker nodes, ref. [37], 2017's throughput severely drops to 290.37 packets/s. In contrast, ref. [20], 2018's throughput was 415.635 packets/s, and our protocol maintains the highest throughput of 441.145 packets/s. According to the results, SecE-V2X outperforms the competing approaches in terms of throughput, confirming that our proposed protocol is indeed effective against Jellyfish reordering attacks.

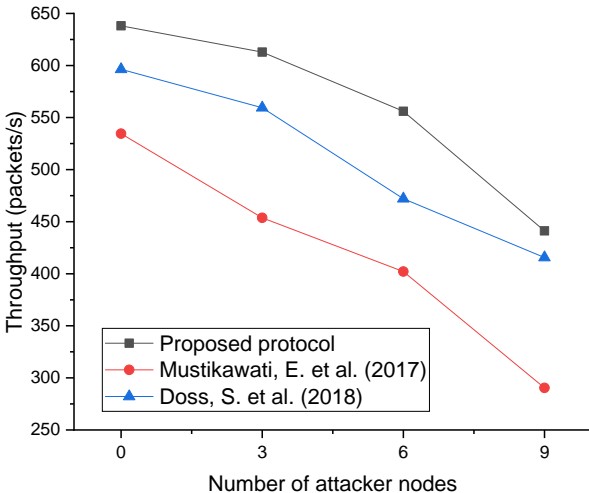

**Figure 18.** Nodes' data throughput under Jellyfish reordering attacks [20,37].

## 6. Conclusions

In this work, we presented an innovative protocol for preventing Jellyfish attacks in V2X, called secure and efficient routing protocol for Jellyfish attack prevention in vehicle-to-everything networks (SecE-V2X). The developed mechanism ensures secure routing using trusted nodes and offers the successful prevention of different Jellyfish attack categories (Jellyfish reorder attacks, Jellyfish periodic drop attacks, and Jellyfish delay variance attacks) with a single solution, despite the fact that no protocol can satisfy all of the requirements of a secure system. In the SecE-V2X technique, the communication behavior between nodes, which is expressed in terms of the packet loss rate, delay, packet reordering, and confidence of a node, is used to compute the overall Honesty value of a node. By choosing the trusted next forwarding node with the highest Honesty value from a current node's neighbors, SecE-V2X ensures secure routing without attacker involvement. With OMNET++ and SUMO simulators, we conducted simulations in an urban environment. In networks of 100 nodes, our protocol was tested against all three Jellyfish attack types. The performance result is compared with two secured routing approaches—[37], 2017, and [20], 2018—in terms of three performance metrics: the packet delivery ratio, end-to-end delay, and throughput. In a scenario without attacks, the results state that SecE-V2X is better in terms of the PDR, with almost 1.54% of [20], 2018, and 3.74% of [37], 2017. SecE-V2X is also better in terms of end-to-end delay (almost 0.31 ms with regard to [20], 2018, and 1.84 ms with regard to [37], 2017). The proposed SecE-V2X maintains a better throughput of almost 41.82 packets/s with regard to [20], 2018, and 103.6 packets/s with regard to [37], 2017. In the presence of six Jellyfish attacker nodes, the results demonstrate that SecE-V2X is better in terms of the PDR, with a PDR of almost 90.50% compared to 79.18% of [20], 2018, and 68.73% of [37], 2017. With a ratio of 8.71 ms, SecE-V2X, with the lowest delay, outperforms [37], 2017 (17.21 ms), and [20], 2018 (11.48 ms), in terms of end-to-end delay. In terms of throughput, SecE-V2X retains the best throughput of 556.02 packets/s, whereas [20], 2018, has a throughput of 472.08 packets/s and [37], 2017, has a throughput of 377.43 packets/s. The results demonstrate that SecE-V2X maintains better performance than that of the competing approaches and works effectively even in the presence of Jellyfish nodes.

With regard to future work, we plan to evaluate and validate our work on top of a realistic dataset of vehicles' mobility.

**Author Contributions:** The major contributions of all the authors are summarized as: Conceptualization, M.B.; methodology, B.B. and N.A.; software, M.B.; validation, A.K. and B.B.; formal analysis, M.B.; investigation, A.K. and B.B.; resources, N.A.; data curation, M.B. and A.K.; writing—original draft preparation, M.B., A.K., B.B. and N.A.; writing—review and editing, M.B. and N.A.; visualization, M.B.; supervision, A.K.; project administration, A.K. All authors have read and agreed to the published version of the manuscript.

**Funding:** This research received no external funding.

**Data Availability Statement:** The data that support the findings of this study are available from the corresponding author upon reasonable request.

**Conflicts of Interest:** The authors declare no potential conflict of interests.

## Abbreviations

| Acronym | Description |
| --- | --- |
| ACKs | Acknowledgments |
| AODV | Ad hoc on-demand distance vector |
| DDoS | Distributed denial of service |
| DoS | Denial of service |
| DPR | Dropped packet ratio |
| DSR | Dynamic source routing |
| DSRC | Dedicated short-range communications |
| DTD | Direct trust-based detection |
| EED | End-to-end delay |
| FIFO | First in, first out |
| GPS | Geographic positioning system |
| GPSR | Greedy perimeter stateless routing protocol |
| IDS | Intrusion detection system |
| ITS | Intelligent transportation system |
| MANET | Mobile ad hoc networks |
| NRL | Normalized routing load |
| NS-2 | Network simulator 2 |
| OLSR | Optimized link state routing protocol |
| OMNET++ | Objective modular network testbed in C++ |
| PDR | Packet delivery ratio |
| QoS | Quality of service |
| RREP | Route reply |
| RREQ | Route request |
| RTO | Retransmission timeout |
| SUMO | Simulator for urban mobility |
| SVM | Support vector machine |
| TCP | Transmission control protocol |
| V2I | Vehicle-to-road infrastructure |
| V2V | Vehicle-to-vehicle |
| V2X | Vehicle-to-everything |
| VANETs | Vehicular ad hoc networks |
| Veins | Vehicles in network simulation |

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
