# Peer review of "Towards Mitigating Jellyfish Attacks Based on Honesty Metrics in V2X Autonomous Networks"

_applsci, doi:10.3390/app13074591_

Round 1
Reviewer 1 Report
The paper presents Towards Mitigating JellyFish Attacks based on Honesty Metric 2 in V2X Autonomous Networks, and the manuscript is well written. Overall, the authors have done an exemplary job in preparing the manuscript. However, I suggest further comments on specific aspects of the protocol or the manuscript.
"The reason and motivation for the use of using honesty criterion mechanism to address the trusted paths is not very clear and it has to be justified. Make sure the abbreviations are mentioned the first time they are used. Your proposal scheme has been evaluated using OMNET++ simulations, it would be appreciated if you insert actual Simulation Scenario, where readers can visualize the original experiment. Add an insightful discussion of how this approach can help in real world scenario. it can also be interesting to discuss why the results are significant and how they add to existing knowledge compare to current techniques. The limitation of proposed approach are not clear. F.6 Flowchart has problem with flow please correct it. The complexity of the all proposed algorithms should be discussed. The (Conclusion) needs some "main" numerical values of the obtained results."
Reviewer 2 Report
In this article study, the authors designed a new protocol that analyzes the behavior of each node in the network and calculates different metrics and selects reliable paths for data transmission to the targeted network. The OMNET++ simulator was used to evaluate the overall performance of the proposed protocol. The study is an important and critical application. The points that leave a question mark about the study and need improvement are listed below.
1- Real-time screen outputs and graphics obtained from the application moment are not presented in the article. Screenshots of real-time parameters made on the OmNet++ simulator used in the study should be presented.
2) The processes in the flow chart given in Figure 6 should be detailed for the protocol proposed in the study. Parameter selection, network flow, duration, time, layers, etc. Also, an algorithm must have a beginning and an end. However, there are missing points on the flowchart.
3) The simulation results on the OmNeT++ platform should be compared in a table according to different throughput values.
4) The organization of article should be reorganized and it should be traceable.
5) There are intermediate paragraphs in the text of the article that disrupt the integrity of meaning. Integrity has not been achieved.
6) The applications made for each different attack type should be given on a separate case and the validity and reliability tests should be examined.
Round 2
Reviewer 1 Report
The authors have addressed all of the necessary changes.
